Plant morphological traits and ecological stoichiometry in disturbed vs. conserved forests

Guan Xiaoyin 1
Xia Haojun 1
Li Shuming 1
Yu Shuisheng 2
Zheng Zihong 2
Liu Julian 2
Liu Libin 1 2 liulibin@zjnu.cn
1 College of Life Sciences, Zhejiang Normal University , Jinhua , China
2 The Administration Center of Zhejiang Jiulong Mountain National Nature Reserve , Lishui , China
Daehler Curtis
Electronic publication date: 2025 Oct 31
Publication date: 2025
Volume: 13
Electronic Location ID: e20301
Received 2025 Jun 3; Accepted 2025 Oct 6
Copyright: © 2025 Guan et al.
Copyright year: 2025
Copyright holder: Guan et al.
License: This is an open access article distributed under the terms of the Creative Commons Attribution License, which permits unrestricted use, distribution, reproduction and adaptation in any medium and for any purpose provided that it is properly attributed. For attribution, the original author(s), title, publication source (PeerJ) and either DOI or URL of the article must be cited.
License URL: https://creativecommons.org/licenses/by/4.0/

Keywords: Plant functional traits, Anthropogenic disturbances, Environments, Adaptation strategy, Subtropical forests

Funding: Natural Science Foundation of Zhejiang Province LQ20C030003 This work was supported by the Natural Science Foundation of Zhejiang Province (LQ20C030003). The funders had no role in study design, data collection and analysis, decision to publish, or preparation of the manuscript.

==============================
Global forests are currently facing significant anthropogenic disturbances. Previous research on plant functional traits has predominantly focused on relatively intact forests, often overlooking those that have experienced such disturbances. This oversight has lead to a scarcity of relevant data regarding disturbed forests in the global and Chinese plant functional trait databases, thereby limiting our understanding of the life history strategies employed by plants inhabiting these altered environments. This study presents data on 12 morphological traits and 24 ecological stoichiometry traits for 62 common species in disturbed forests and 43 species in conserved forests in East China. We analyzed the variability characteristics of these functional traits, explored functional trait differences between disturbed and conserved forests, and examined relationships among various functional traits to investigate disparities in life history strategies between the two forest types. The results indicated that the variability of plant functional traits was generally lower in disturbed forests compared to conserved ones. Most functional traits exhibited significant differences between the two forest types (P < 0.05). Additionally, stronger correlations among functional traits were noted in disturbed forests. From a functional trait perspective, plants in disturbed forests displayed high trait correlations and formed trait combinations indicative of a resource conservative strategy characterized by low specific leaf area, high dry matter content and tissue density across leaves, twigs, barks and stems; alongside heightened carbon investment but reduced al locations for nitrogen and phosphorus. A comprehensive investigation of plant functional traits in both disturbed and conserved forests will enrich the global and Chinese trait databases, providing insights into how forest plants adapt to disturbances and informing ecological restoration in degraded areas.

Introduction

Forest disturbance refers to the reduction or complete loss of forest canopy cover and biomass reservoirs, resulting from both natural and anthropogenic factors (De Marzo et al., 2023; Requena Suarez et al., 2023; Smith et al., 2023; He, Hong & Zhu, 2024). Global forests are currently facing substantial disturbances driven by human activities as well as natural events (Walker, 2024; Acil et al., 2025). Exploring the mechanisms by which disturbances affect forest ecosystems is a crucial area of inquiry within the domains of conservation ecology and global change ecology. Traditional methodologies that focus on analyzing species composition and community dynamics have become increasingly inadequate for addressing contemporary challenges. Consequently, it is imperative to investigate alternative approaches to quantify and predict the effects of disturbances on biodiversity, structure, processes, and functions within forest ecosystems (Liu & Ma, 2015).

The research on plant functional traits has experienced substantial growth over the past two decades, infusing new vitality into ecological studies amidst global forest disturbances (Verheyen et al., 2003; Vandewalle et al., 2010; Loto & Bravo, 2020; Liu et al., 2024). As intrinsic physiological and extrinsic morphological characteristics shaped by the interactions between plants and their living environments, plant functional traits not only indicate how plants respond to and adapt to external conditions but also reflect the functional attributes of ecosystems (Reich et al., 2003; Violle et al., 2007; Díaz et al., 2016; Blondeel et al., 2020). For example, plant morphological traits such as maximum tree height, specific leaf area (SLA), and wood density are critical in determining plant distribution patterns and adaptations in life history strategies (Westoby, 1998; Poorter et al., 2010; Liu et al., 2022); Furthermore, the carbon, nitrogen, and phosphorus contents—along with their ratios—found in plant leaves, branches, trunks, and barks are essential for elucidating ecological processes including energy flow, material cycling, and nutrient limitation within ecosystems (Elser et al., 2000; Elser, Acquisti & Kumar, 2011; Liu et al., 2022). Consequently, plant functional traits serve as an effective integration of individual plants, environmental factors, and the structure, processes, and functions of ecosystems (Koerselman & Meuleman, 1996; Mcgill et al., 2006; Meng, Ni & Wang, 2007; Liu & Ma, 2015).

In previous studies, scientists have primarily focused on the functional traits of plants in natural forests. However, there has been limited investigation into the effects of various anthropogenic disturbances—such as fire, grazing, biological invasion, and land use change—on plant functional traits (Verheyen et al., 2003; Pausas et al., 2004; Lamarque, Delzon & Lortie, 2011; Liu & Ma, 2015; Loto & Bravo, 2020). The current global and Chinese plant functional trait databases primarily comprise data derived from intact forests, with considerably less representation from anthropogenically disturbed forests (Kattge et al., 2011, 2020; Martin et al., 2018; Wang et al., 2018, 2022). Anthropogenic disturbances modify environmental conditions in forests. Plants growing in forests that exhibit varying disturbance histories across diverse environments may display different characteristics in their functional traits and adopt distinct life history strategies (Blair et al., 2016; Poorter et al., 2018; Loto & Bravo, 2020; Liu et al., 2024). Consequently, this disparity in plant functional traits between disturbed and conserved forests inevitably affects our comprehensive understanding of the global plant trait spectrum, introduces biases into global and regional plant trait mapping, and impedes our capacity to accurately assess the structure and functions of anthropogenically disturbed forests as well as their potential for ecological restoration (Díaz et al., 2016; Butler et al., 2017; Liu et al., 2024).

In this study, we established a comprehensive database of plant functional traits that encompasses 12 morphological traits and 24 ecological stoichiometry traits related to leaves, twigs, barks and stems from 62 common species in anthropogenically disturbed forests and 43 common species in conserved forests in East China. Utilizing this database, we analyzed the correlations among these 36 functional traits and compared the trait differences between plants found in disturbed vs. conserved forests to further investigate the life history strategies of plants from disturbed environments. Accordingly, two predictions were formulated: (1) plant functional traits and their correlations differ significantly between disturbed and conserved forests. (2) Disturbed forests exhibit high trait correlations and form specific trait combinations characterized by low SLA, nitrogen and phosphorus contents across leaves, twigs, barks and stems, alongside high dry matter content, tissue density and carbon content across these same structures—indicative of a resource conservative strategy. This study not only contributes valuable data to the global and Chinese trait databases but also provides a theoretical foundation for exploring the mechanisms underlying forest degradation subjected to anthropogenic disturbances, as well as practical guidance for forest restoration and reconstruction efforts in East China.

Materials and Methods

Study area

The Jiulong Mountain National Nature Reserve (118°49′–118°55′E, 28°19′–28°24′N) is located in Suichang County, Zhejiang Province, East China. This reserve encompasses a contiguous area that spans 8.8 km from east to west and 10.5 km from north to south, covering a total area of 55.25 km2 (Fig. 1). Its strategic location at the confluence of Zhejiang, Fujian, and Jiangxi provinces elevates it to the status of one of the 25 global priority areas and one of the 35 key regions in China dedicated to biodiversity conservation (Myers et al., 2000; Ministry of Ecology and Environment of the People’s Republic of China (MEEC), 2011). The region exhibits a mid-subtropical humid monsoon climate. The annual precipitation measures 1,856 mm, with a relative humidity of 80%. The total duration of sunshine throughout the year is approximately 1,925 h. The average annual temperature stands at 16.2 °C, while the recorded extreme high temperature reaches 37.0 °C and the extreme low temperature drops to −10.5 °C (Liu et al., 2025). The geological foundation is primarily composed of weathering products derived from acidic igneous rocks. The predominant soil types include mountainous red-yellow soil and paddy soil, with a particular emphasis on the former (Li, 2008). Dominant vegetation within the reserve consists of zonal evergreen broadleaf forest, with various other types interspersed, including mixed evergreen-deciduous broadleaf forest, deciduous broadleaf forest, coniferous forest, mixed coniferous-broadleaf forest, and bamboo forest (Liu et al., 2024).

Figure 1 Locations of sampling plots within the Jiulong Mountain National Nature Reserve and its surrounding protective buffer zone in East China.

History of sampling sites and vegetation survey

Surveys and data collection were carried out in the Jiulong Mountain National Nature Reserve and its surrounding protective buffer zone (Fig. 1). The reserve is home to largely intact native plant communities and serves as a refuge for numerous endangered and rare species, attributed to its isolated location and limited transportation infrastructure. Initially designated as a provincial-level protected area in 1983, it was elevated to national status in 2003. Over the past four decades, the management authority of the reserve has implemented stringent regulations against disruptive activities such as fire, timber harvesting, fuelwood collection, and grazing. While similar robust fire prevention policies are enforced in the surrounding protective buffer zone, intermediate anthropogenic disturbances continue to occur. In natural broadleaf forests—including evergreen broadleaf forests, mixed evergreen-deciduous broadleaf forests, and deciduous broadleaf forests—fuelwood collection and grazing occasionally take place. Meanwhile, in mixed coniferous-broadleaf forests, coniferous forests, and bamboo forests, timber harvesting has occurred alongside fuelwood collection and grazing.

Following a comprehensive vegetation survey, we selected 22 plots within the reserve (representing conserved forests) and 44 plots in the surrounding protective buffer zone (representing disturbed forests). The latter exhibited nearly uniform intensity and frequency of fuelwood collection and grazing across all forest types, as well as timber harvesting activities that displayed similar patterns in mixed coniferous-broadleaf forests, coniferous forests, and bamboo forests (Table S1). This selection encompassed all common forest types (Fig. 1). The distance between plots of the same forest type was maintained at greater than 300 m. Each plot measured 20 m × 20 m. For each plot, key attributes were documented, including GPS location, elevation, slope, aspect, and coverage of outcrops. All woody plants with a diameter at breast height (D) ≥ 5 cm were identified; their diameter at breast height (DBH), height, and crown width were measured accordingly.

Functional trait measurements

Based on the plot survey data, a total of 62 common species—defined as those with at least five individuals recorded in either disturbed or conserved forests—in disturbed forests and 43 species in conserved forests were selected for further plant functional trait measurements. For each species, five robust and dominant individuals were chosen. From every tree, four branches were collected an extendable tree pruner, with each branch sources from distinct locations on the sunlit portion of the crown. For each branch, either five fully developed broad leaves or ten mature needle leaves were harvested, along with one 2–3-year-old twig approximately 20 cm in length. In total, 20 or 40 leaves and four twigs were sampled from each individual. Additionally, three bark samples (it should noted that insufficient bark samples were collected from Phyllostachys edulis, Rhododendron ovatum, Eurya muricata, and Itea omeiensis due to the challenges associated with sampling the bark of these species) and three stem samples (insufficient stem samples were also collected from Trachycarpus fortunei due to difficulities encountered while sampling this species’ stem) were procured near the D position from each individual using a sickle and an increment borer. All samples were aseptically transferred to sterile dry bags to preserve integrity for subsequent laboratory analyses.

Five leaves were meticulously stacked together, ensuring that the main veins were excluded from the measurement. The thickness of the stacked leaves was measured using a vernier caliper with a precision of 0.01 mm. The average thickness of these collected leaves for an individual was used to represent the leaf thickness (LT) for that individual. The areas of all leaves were scanned utilizing the WinFOLIA multipurpose leaf area meter (Regent Instruments, Quebec, Canada). The leaf area (LA) for each individual was calculated by dividing the total area of all leaves by their total number. The fresh leaf mass of an individual was determined using an electronic balance with a precision of 0.001 g. Subsequently, the leaves were dried at 80 °C for 48 h until they reached a constant weight, after which dry mass was recorded. Leaf volume was calculated as the product of LA and LT. Specific leaf area was determined by dividing LA by dry mass. Leaf dry matter content (LDMC) was assessed by dividing leaf dry mass by fresh mass, while leaf tissue density (LTD) was calculated by dividing leaf dry mass by volume. A vernier caliper was employed to measure the thickness of bark samples, with average thickness derived from three bark samples representing bark thickness (BT) for this individual. Fresh masses of twig, bark and stem from an individual were similarly assessed using an electronic balance. Volume determinations were conducted employing the appropriate drainage method. Subsequently, all twig, bark, and stem samples were dried at 80 °C for 72 h to obtain their respective dry masses. Twig dry matter content (TDMC), bark dry matter content (BDMC), and stem dry matter content (SDMC) for each individual were calculated by dividing average dry masses by corresponding average fresh masses. Additionally, twig tissue density (TTD), bark tissue density (BTD), and stem tissue density (STD) for this individual were determined through division of dry masses by their respective average volumes (Cornelissen et al., 2003; Pérez-Harguindeguy et al., 2013).

Following the measurement of morphological traits, all samples were ground into a fine powder and passed through a 0.2 mm mesh sieve. Leaf, twig, bark, and stem samples were analyzed for total carbon and nitrogen contents (denoted as leaf carbon content (LC), leaf nitrogen content (LN); twig carbon content (TC), twig nitrogen content (TN); bark carbon content (BC), bark nitrogen content (BN); stem carbon content (SC), stem nitrogen content (SN)) using the Vario MACRO Cube (Thermo Scientific, Waltham, MA, USA). Total phosphorus contents (leaf phosphorus (LP), twig phosphorus (TP), bark phosphorus (BP), stem phosphorus (SP)) were determined using the iCAP 6300 ICP-OES Spectrometer Analyzer (Thermo Scientific, Waltham, MA, USA). Subsequently, stoichiometric ratios—including carbon-to-nitrogen ratio, carbon-to-phosphorus ratio, and nitrogen-to-phosphorus ratio—were calculated for each tissue type: leaf (leaf carbon-to-nitrogen ratio (LCN), leaf carbon-to-phosphorus ratio (LCP), leaf nitrogen-to-phosphorus ratio (LNP)), twig (twig carbon-to-nitrogen ratio (TCN), twig carbon-to-phosphorus ratio (TCP), twig nitrogen-to-phosphorus ratio (TNP)), bark (bark carbon-to-nitrogen ratio (BCN), bark carbon-to-phosphorus ratio (BCP), bark nitrogen-to-phosphorus ratio (BNP)), and stem (stem carbon-to-nitrogen ratio (SCN), stem carbon-to-phosphorus ratio (SCP), stem nitrogen-to-phosphorus ratio (SNP)).

Data analysis

For data analysis, statistical procedures were conducted using R software version 4.1.3 (R Core Team, 2018). Coefficients of variation, defined as the standard deviation divided by the mean, were utilized to quantify the variability of plant functional traits. An independent samples t-test was performed to assess differences in plant functional traits between disturbed and conserved forests. Additionally, Pearson correlation analysis was employed to investigate relationships among functional traits.

Results

Morphological traits

There were no significant differences in LT, LA, LTD and BT between plants in disturbed forests and those in conserved forests (P > 0.05). However, the other eight morphological traits exhibited significant differences between the two forest types (P < 0.05) (Fig. 2). Notably, SLA was significantly higher in conserved forests compared to disturbed ones, while the remaining seven morphological traits were found to be significantly higher in disturbed forests than in conserved ones (P < 0.05) (Fig. 2).

Figure 2 Differences in plant morphological traits between disturbed and conserved forests in East China (A–L).

The black dots represent the morphological trait values of all species, while P < 0.05 and P < 0.01 indicate statistically significant differences observed (t-test). Abbreviations: LT, leaf thickness; LA, leaf area; SLA, specific leaf area; LDMC, leaf dry matter content; LTD, leaf tissue density; TDMC, twig dry matter content; TTD, twig tissue density; BT, bark thickness; BDMC, bark dry matter content; BTD, bark tissue density; SDMC, stem dry matter content; STD, stem tissue density.

The coefficients of variation for morphological traits in plants from disturbed forests ranged from 9.97% to 78.81% (mean = 34.14%), while those in plants from conserved forests varied between 10.05% and 93.33% (mean = 37.73%) (Table 1). Morphological traits in plants from conserved forests exhibited greater variability than those in plants from disturbed forests, with the exception of LTD (Table 1). In disturbed forests, BT displayed the highest variation, whereas TDMC showed the lowest variation as indicated by their respective coefficients of variation; conversely, in conserved forests, LA demonstrated the greatest variability while TDMC again presented the least variability as reflected by their corresponding coefficients of variation (Table 1).

Table 1 Plant morphological traits in disturbed vs. conserved forests in East China.

Morphological traits	Disturbed forests	Conserved forests	
Number of species	Mean	Max	Min	Coefficient of variation (%)	Number of species	Mean	Max	Min	Coefficient of variation (%)	
LT (mm)	62	0.4 ± 0.15	1.02	0.10	36.92	43	0.35 ± 0.17	0.91	0.14	47.70	
LA (cm2)	62	20.42 ± 15.72	94.97	0.21	77.00	43	26.71 ± 24.93	154.13	0.19	93.33	
SLA (cm2 g−1)	62	122.11 ± 47.41	256.52	57.29	38.83	43	165 ± 83.76	360.09	41.63	50.77	
LDMC (g g−1)	62	0.41 ± 0.07	0.55	0.23	15.99	43	0.36 ± 0.07	0.49	0.18	19.09	
LTD (g cm−3)	62	0.58 ± 0.35	2.41	0.27	59.23	43	0.57 ± 0.33	1.98	0.13	58.07	
TDMC (g g−1)	62	0.53 ± 0.05	0.65	0.40	9.97	43	0.49 ± 0.05	0.57	0.37	10.05	
TTD (g cm−3)	62	0.55 ± 0.09	0.73	0.36	17.20	43	0.47 ± 0.09	0.68	0.31	19.38	
BT (mm)	61	2.72 ± 2.14	12.86	0.64	78.81	41	3.71 ± 3.02	14.82	0.88	81.31	
BDMC (g g−1)	61	0.54 ± 0.08	0.80	0.31	15.74	41	0.5 ± 0.08	0.67	0.23	15.77	
BTD (g cm−3)	61	0.51 ± 0.12	0.87	0.19	23.16	41	0.43 ± 0.12	0.63	0.16	26.83	
SDMC (g g−1)	62	0.55 ± 0.09	0.71	0.33	15.94	43	0.52 ± 0.07	0.62	0.32	13.24	
STD (g cm−3)	62	0.58 ± 0.12	0.84	0.32	20.91	43	0.53 ± 0.09	0.69	0.31	17.16	
Note:

LT, leaf thickness; LA, leaf area; SLA, specific leaf area; LDMC, leaf dry matter content; LTD, leaf tissue density; TDMC, twig dry matter content; TTD, twig tissue density; BT, bark thickness; BDMC, bark dry matter content; BTD, bark tissue density; SDMC, stem dry matter content; STD, stem tissue density.

Ecological stoichiometry

There were no significant differences in LC, LNP, TCP, TNP, BC, BN, BCN, SC, SN and SCN between plants in disturbed forests and those in conserved forests (P > 0.05). However, the remaining 14 ecological stoichiometry traits exhibited significant differences between the two forest types (P < 0.05) (Fig. 3). Notably, LN, LP, TN, TP, BP and SP were significantly higher in conserved forests compared to disturbed forests. Conversely, TC, LCN, LCP, TCN, BCP, BNP, SCP and SNP were found to be significantly elevated in disturbed forests relative to conserved ones (P < 0.05) (Fig. 3).

Figure 3 Differences in ecological stoichiometry between disturbed and conserved forests in East China (A–X).

The black dots represent the ecological stoichiometry values of all species, while P < 0.05 and P < 0.01 indicate statistically significant differences observed (t-test). Abbreviations: LC, leaf carbon content; LN, leaf nitrogen content; LP, leaf phosphorus content; LCN, leaf carbon-to-nitrogen ratio; LCP, leaf carbon-to-phosphorus ratio; LNP, leaf nitrogen-to-phosphorus ratio; TC, twig carbon content; TN, twig nitrogen content; TP, twig phosphorus content; TCN, twig carbon-to-nitrogen ratio; TCP, twig carbon-to-phosphorus ratio; TNP, twig nitrogen-to-phosphorus ratio; BC, bark carbon content; BN, bark nitrogen content; BP, bark phosphorus content; BCN, bark carbon-to-nitrogen ratio; BCP, bark carbon-to-phosphorus ratio; BNP, bark nitrogen-to-phosphorus ratio; SC, stem carbon content; SN, stem nitrogen content; SP, stem phosphorus content; SCN, stem carbon-to-nitrogen ratio; SCP, stem carbon-to-phosphorus ratio; SNP, stem nitrogen-to-phosphorus ratio.

The coefficients of variation for ecological stoichiometry in plants from disturbed forests ranged from 1.63% to 45.27% (mean = 25.11%), whereas those in plants from conserved forests varied between 1.99% and 57.81% (mean = 29.23%) (Table 2). In most instances, ecological stoichiometry in plants from conserved forests exhibited greater variability compared to those in plants from disturbed forests, with the exceptions being LNP, TN, TCN, BN, SCP (Table 2). In disturbed forests, BN displayed the highest degree of variation, while TC showed the lowest level of variation as indicated by their respective coefficients of variation; conversely, in conserved forests, TNP demonstrated the greatest variability while SC exhibited the least variability as reflected by their corresponding coefficients of variation (Table 2).

Table 2 Plant ecological stoichiometry in disturbed vs. conserved forests in East China.

Ecological stoichiometry	Disturbed forests	Conserved forests	
Number of species	Mean	Max	Min	Coefficient of variation (%)	Number of species	Mean	Max	Min	Coefficient of variation (%)	
LC (mg g−1)	62	467.59 ± 22.6	507.26	409.62	4.83	43	461.26 ± 31.79	505.36	341.51	6.89	
LN (mg g−1)	62	14.73 ± 4.36	32.24	7.01	29.60	43	17.41 ± 5.58	29.44	8.31	32.03	
LP (mg g−1)	62	0.94 ± 0.24	1.46	0.60	25.59	43	1.09 ± 0.33	1.91	0.49	30.23	
LCN	62	34.35 ± 9.91	66.60	15.04	28.83	43	29.39 ± 10.08	57.98	15.48	34.31	
LCP	62	529.96 ± 134.97	796.02	304.92	25.47	43	470.58 ± 171.12	973.46	226.40	36.36	
LNP	62	15.76 ± 2.8	25.98	10.57	17.74	43	16.11 ± 2.11	22.97	12.65	13.12	
TC (mg g−1)	62	468.32 ± 7.63	486.38	452.94	1.63	43	451.8 ± 17.84	489.05	413.54	3.95	
TN (mg g−1)	62	3.39 ± 1.05	6.58	1.32	30.87	43	3.92 ± 1.18	6.62	1.72	30.22	
TP (mg g−1)	62	0.54 ± 0.13	0.94	0.20	23.48	43	0.62 ± 0.24	1.42	0.23	38.61	
TCN	62	153.46 ± 57.33	363.95	70.61	37.36	43	127.7 ± 46.01	274.46	68.78	36.03	
TCP	62	931.53 ± 290.84	2,441.45	491.85	31.22	43	845.38 ± 359.36	2,040.20	305.81	42.51	
TNP	62	6.4 ± 1.71	12.31	3.58	26.67	43	7.25 ± 4.19	24.15	3.03	57.81	
BC (mg g−1)	61	464.22 ± 27.63	517.39	399.11	5.95	41	452.84 ± 33.29	524.17	392.37	7.35	
BN (mg g−1)	61	6.52 ± 2.95	22.60	2.30	45.27	41	6.11 ± 2.38	12.69	2.31	38.99	
BP (mg g−1)	61	0.47 ± 0.11	0.97	0.22	23.67	41	0.59 ± 0.16	1.21	0.26	27.39	
BCN	61	80.71 ± 29.43	224.95	20.64	36.46	41	84.66 ± 33.3	219.06	33.77	39.33	
BCP	61	1,044.7 ± 299.36	2,371.78	466.18	28.65	41	824.77 ± 239.48	1,657.55	356.33	29.04	
BNP	61	14.01 ± 5.33	39.07	7.56	38.07	41	10.89 ± 5.04	33.94	3.61	46.29	
SC (mg g−1)	62	466.8 ± 7.83	482.35	446.53	1.68	43	463.67 ± 9.23	482.89	439.40	1.99	
SN (mg g−1)	62	2.11 ± 0.6	4.11	1.09	28.71	43	2.06 ± 0.69	3.49	0.99	33.55	
SP (mg g−1)	62	0.26 ± 0.07	0.49	0.14	28.27	43	0.47 ± 0.14	1.08	0.30	28.81	
SCN	62	239.76 ± 67.87	440.89	108.64	28.31	43	252.47 ± 87.89	475.66	129.75	34.81	
SCP	62	1,947.39 ± 523.31	3,371.80	954.26	26.87	43	1,041.35 ± 241.85	1,549.37	417.53	23.22	
SNP	62	8.45 ± 2.31	14.97	4.14	27.33	43	4.4 ± 1.26	8.70	2.41	28.61	
Note:

Abbreviations: LC, leaf carbon content; LN, leaf nitrogen content; LP, leaf phosphorus content; LCN, leaf carbon-to-nitrogen ratio; LCP, leaf carbon-to-phosphorus ratio; LNP, leaf nitrogen-to-phosphorus ratio; TC, twig carbon content; TN, twig nitrogen content; TP, twig phosphorus content; TCN, twig carbon-to-nitrogen ratio; TCP, twig carbon-to-phosphorus ratio; TNP, twig nitrogen-to-phosphorus ratio; BC, bark carbon content; BN, bark nitrogen content; BP, bark phosphorus content; BCN, bark carbon-to-nitrogen ratio; BCP, bark carbon-to-phosphorus ratio; BNP, bark nitrogen-to-phosphorus ratio; SC, stem carbon content; SN, stem nitrogen content; SP, stem phosphorus content; SCN, stem carbon-to-nitrogen ratio; SCP, stem carbon-to-phosphorus ratio; SNP, stem nitrogen-to-phosphorus ratio.

Functional trait relationships

Pearson correlation analysis revealed that a considerable number of plant functional traits exhibited significant correlations with one another in both disturbed and conserved forests. Among the 630 pairwise trait combinations involving the 36 traits, 292 combinations (46.35% of the total) demonstrated significant correlations in disturbed forests, while 274 combinations (43.49%) showed significant correlations in conserved forests (P < 0.05) (Fig. 4; Tables S2 and S3).

Figure 4 Relationships among plant functional traits in disturbed forest (A) and conserved forest (B) in East China.

Asterisks (*, ** and ***) indicate significant correlations identified through Pearson correlation analysis (P < 0.05/0.01/0.001), while the symbol × denotes the absence of significant correlations (Pearson correlation analysis, P > 0.05). Abbreviations: LT, leaf thickness; LA, leaf area; SLA, specific leaf area; LDMC, leaf dry matter content; LTD, leaf tissue density; TDMC, twig dry matter content; TTD, twig tissue density; BT, bark thickness; BDMC, bark dry matter content; BTD, bark tissue density; SDMC, stem dry matter content; STD, stem tissue density. LC, leaf carbon content; LN, leaf nitrogen content; LP, leaf phosphorus content; LCN, leaf carbon-to-nitrogen ratio; LCP, leaf carbon-to-phosphorus ratio; LNP, leaf nitrogen-to-phosphorus ratio; TC, twig carbon content; TN, twig nitrogen content; TP, twig phosphorus content; TCN, twig carbon-to-nitrogen ratio; TCP, twig carbon-to-phosphorus ratio; TNP, twig nitrogen-to-phosphorus ratio; BC, bark carbon content; BN, bark nitrogen content; BP, bark phosphorus content; BCN, bark carbon-to-nitrogen ratio; BCP, bark carbon-to-phosphorus ratio; BNP, bark nitrogen-to-phosphorus ratio; SC, stem carbon content; SN, stem nitrogen content; SP, stem phosphorus content; SCN, stem carbon-to-nitrogen ratio; SCP, stem carbon-to-phosphorus ratio; SNP, stem nitrogen-to-phosphorus ratio.

Discussion

Plant functional traits have been demonstrated to be an effective and reliable approach for investigating a range of pressing issues in ecology (Liu & Ma, 2015; He et al., 2023). Consequently, plant ecologists have devoted considerable attention to the study of plant functional traits, leading to intensive measurements of various types across forests worldwide (Chave et al., 2009; Auger & Shipley, 2013; Li et al., 2022; Liu et al., 2024). However, most studies have predominantly focused on leaf morphological traits such as LT, LA, SLA, LDMC and LTD (Heilmeier, 2019; Liu et al., 2023). This focus is largely due to leaves being the organs with the largest surface area exposed to the external environment, their heightened sensitivity to numerous environmental changes, and the relative ease with which these morphological traits can be observed and measured (Wright et al., 2004; Pérez-Ramos et al., 2012; Díaz et al., 2016; Wu et al., 2023). In contrast, the characteristics of twig, bark and stem morphological traits, as well as the C:N:P ecological stoichiometry associated with these plant organs remain relatively underexplored (Liu et al., 2022). The database established in this study encompasses 12 morphological traits and 24 ecological stoichiometry traits across numerous species in forests (Table S4). This research enhances the analysis of synergies and trade-offs associated with the functional traits of leaves, twigs, stems, and barks in East China.

Plant functional trait data are frequently employed to investigate plant life history strategies (Wright et al., 2004; Díaz et al., 2016). Previous studies utilizing leaf trait data from the global and Chinese trait databases have shown that certain plants exhibit high SLA, elevated LN and LP, increased photosynthetic rates, and shorter leaf lifespans, indicative of a resource acquisitive strategy. Conversely, other plants demonstrate high LDMC, LTD and LC, and long leaf lifespans, reflecting a resource conservative strategy (Chen & Xu, 2014). In this study, the disturbed forests represent relatively fragile ecosystems marked by poor soil nutrients and fluctuating soil temperature and humidity. Plants in such forests adopt a resource conservative strategy characterized by a combination of traits including low SLA, high DMC and TD across leaves, twigs, barks and stems. This is accompanied by increased carbon investment but reduced allocations for nitrogen and phosphorus. In contrast, plants in conserved forests exhibit a resource acquisitive strategy with opposing trait combinations. Furthermore, stronger correlations among traits were noted in disturbed forests; this serves as additional evidence that these plants allocate more resources to resist disturbances. Our results support our predictions and align with previous studies. Moreover, certain traits exhibited significant differences while others did not; this variation can be attributed to the differing ecological sensitivities of these traits in response to disturbances (Liu et al., 2024).

In addition to examining life history strategies, the database established in this study can also be employed to explore community species composition and dynamics, functional diversity, as well as assess the impacts of environmental changes due to disturbances on ecosystems (Petchey & Gaston, 2002; Bernhardt-Roemermann et al., 2011; Liu et al., 2025).

However, it is essential to emphasize that the current database includes only the morphological traits of leaves, twigs, barks, and stems, as well as the ecological stoichiometry of plants found in both disturbed and conserved forests. Further investigation and supplementation in future studies will be necessary to explore additional functional traits, such as root morphological characteristics and ecological stoichiometry, along with leaf photosynthesis and shade tolerance of plants from other disturbed and conserved ecosystems.

The forests in East China have consistently faced significant degradation challenges due to intensive anthropogenic activities. Currently, the restoration of degraded vegetation—such as grasslands, shrublands, and secondary forests—has emerged as a critical ecological issue in this region (Liu et al., 2023, 2024). Typically, enhancing species diversity and increasing carbon stock are employed as key indicators for evaluating the effectiveness of these restoration initiatives (Yang et al., 2022). Plant functional traits offer an alternative approach for predicting the outcomes of restoration efforts and assessing the potential for local vegetation recovery (Sandel, Corbin & Krupa, 2011; Hedberg et al., 2013). In this study, we observed that numerous plant functional traits exhibited significant differences between disturbed and conserved forests, indicating that utilizing plant functional trait data to reflect forest restoration efforts is indeed feasible. Therefore, a comprehensive exploration of plant functional trait characteristic and trait-based life history strategies within both disturbed and conserved forest ecosystems is essential not only for contributing data to the global and Chinese trait databases but also for benchmarking the next-generation vegetation models (Jin et al., 2023). Furthermore, it provides valuable insights into how forest plants adapt to disturbances and informs ecological restoration in degraded areas of East China.

Conclusions

In this study, we established a comprehensive database encompassing 36 plant functional traits across 62 common species in disturbed forests and 43 species in conserved forests. Utilizing this database, we investigated the life history strategies of plants in disturbed forests. Our findings revealed that plants in disturbed forests displayed strong trait correlations and formed trait combinations indicative of a resource conservative strategy characterized by low SLA, high dry matter content and tissue density across leaves, twigs, barks and stems; alongside heightened carbon investment but reduced allocations for nitrogen and phosphorus. Further investigation and supplementary research will be essential to explore additional functional traits of plants from both disturbed and conserved ecosystems.

Supplemental Information

Supplemental Information 1 Plant morphological traits and ecological stoichiometry in disturbed versus conserved forests in East China.

Abbreviations: LT, leaf thickness; LA, leaf area; SLA, specific leaf area; LDMC, leaf dry matter content; LTD, leaf tissue density; TDMC, twig dry matter content; TTD, twig tissue density; BT, bark thickness; BDMC, bark dry matter content; BTD, bark tissue density; SDMC, stem dry matter content; STD, stem tissue density; LC, leaf carbon contents; LN, leaf nitrogen contents; LP, leaf phosphorus contents; LCN, leaf carbon-to-nitrogen ratios; LCP, leaf carbon-to-phosphorus ratios; LNP, leaf nitrogen-to-phosphorus ration; TC, twig carbon contents; TN, twig nitrogen contents; TP, twig phosphorus contents; TCN, twig carbon-to-nitrogen ratios; TCP, twig carbon-to-phosphorus ratios; TNP, twig nitrogen-to-phosphorus ration; BC, bark carbon contents; BN, bark nitrogen contents; BP, bark phosphorus contents; BCN, bark carbon-to-nitrogen ratios; BCP, bark carbon-to-phosphorus ratios; BNP, bark nitrogen-to-phosphorus ration; SC, stem carbon contents; SN, stem nitrogen contents; SP, stem phosphorus contents; SCN, stem carbon-to-nitrogen ratios; SCP, stem carbon-to-phosphorus ratios; SNP, stem nitrogen-to-phosphorus ration.

Additional Information and Declarations

Competing Interests

The authors declare that they have no competing interests.

Author Contributions

Xiaoyin Guan performed the experiments, analyzed the data, prepared figures and/or tables, authored or reviewed drafts of the article, and approved the final draft.

Haojun Xia performed the experiments, analyzed the data, prepared figures and/or tables, and approved the final draft.

Shuming Li analyzed the data, prepared figures and/or tables, and approved the final draft.

Shuisheng Yu performed the experiments, authored or reviewed drafts of the article, and approved the final draft.

Zihong Zheng performed the experiments, authored or reviewed drafts of the article, and approved the final draft.

Julian Liu performed the experiments, authored or reviewed drafts of the article, and approved the final draft.

Libin Liu conceived and designed the experiments, performed the experiments, analyzed the data, prepared figures and/or tables, authored or reviewed drafts of the article, and approved the final draft.

Data Availability

The following information was supplied regarding data availability:

The raw measurements are available in the Supplemental File.

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
