# Peer review of "Plant morphological traits and ecological stoichiometry in disturbed vs. conserved forests"

_PeerJ, doi:10.7717/peerj.20301_

## Round 0.1 · original submission · Major Revisions

The reviewers identified various questions and revisions that should be addressed to improve the manuscript. One crucial need is to clearly describe how "disturbed" forest was objectively identified and how "representative disturbed forest" plots were selected. Did all of the disturbed forest plots experience the same disturbances on the same time scale? If there were different types of disturbances in different plots, how was representation among disturbance types decided when selecting the plots? If plots had different types or recency of disturbances, were there trends in plant traits associated with different types or recency of disturbances? I think it would be valuable to include a table (as supplementary material) that provides basic information on each plot (GPS coordinate, most abundant tree(s), canopy cover if available or other environmental information if available, and type of disturbance(s) for which you found evidence in each plot. Your approach and reasoning for selecting specific plots (disturbed or not) should be clear to readers.

**PeerJ Staff Note**: Please ensure that all review, editorial, and staff comments are addressed in a response letter and that any edits or clarifications mentioned in the letter are also inserted into the revised manuscript where appropriate.

**Language Note**: The review process has identified that the English language must be improved. PeerJ can provide language editing services - please contact us at [email protected] for pricing (be sure to provide your manuscript number and title). Alternatively, you should make your own arrangements to improve the language quality and provide details in your response letter. – PeerJ Staff

Reviewer 1 ·

Basic reporting

The manuscript Plant Morphological traits and ecological stoichiometry in disturbed versus conserved forests raises the analysis of a relevant issue on the variation of functional traits in disturbed and conserved environments. The manuscript is well written; however, in the introduction it does not offer a broad and solid review that justifies the objective of the study, although as the authors point out the topic of functional trait variation in degraded environments is incipient in many ecosystems, there is enough literature to establish more solid research questions. Therefore, a more in-depth review of this topic should be undertaken in this manuscript. The methodological approach is weak and limits the level of analysis and discussion.

Experimental design

The experimental design of the study is deficient; there are no specific criteria (history of use of each sampling site, conservation status, floristic composition and vegetation structure, etc.) to categorize the two forest conditions analyzed (conserved and disturbed). This is very relevant because different studies have shown that the successional status of communities determines the composition of functional traits. Without an objective categorization of the conservation status of the communities (conserved or disturbed) it is not possible to determine that the patterns recorded are associated with the level of disturbance. Like the conservation status of the communities, there is no objective criterion (e.g. abundance) to determine what is meant by common species. Although the authors indicate that in their sampling, they considered only individuals with a dbh > 5 cm, measuring height and cover, they do not show any structural analysis that would serve as a basis for determining which are the structurally relevant species, which will present the functional responses in each state of conservation of the communities.

Validity of the findings

The deficiencies in the experimental design affect the analytical level of the results, which only provide a general description of the central tendencies and levels of variation of the functional traits analyzed. As a result, the discussion does not explain the possible causes of the variation patterns recorded for the functional responses analyzed and their possible effect on the ecosystemic processes. The explanations offered by the authors do not allow us to understand how disturbances affect the functional responses of plants or which are the dominant responses in each forest conservation condition.

Additional comments

The manuscript can be improved by further reviewing the literature on functional trait changes in degraded environments to establish a more solid objective and hypothesis. In addition, they should provide an objective characterization of the conservation status of the plant communities studied, as well as base the selection criteria for species selection on their structural contribution to the communities. Modify the criteria for data analysis, seeking to determine which are the most important functional traits that explain the variation between communities (e.g. multivariate analysis) and the functional relationships between. The discussion should attempt to explain which environmental factors related to conservation status influence plant functional responses.

Reviewer 2 ·

Basic reporting

The manuscript meets the requirements in this area. The language is appropriately used (although there are several typos, see below), and the relevant literature is cited. The figures and tables are clear and easy to understand. At first, I thought Figure 3 was too busy, but it grew on me. Figures 1 and 2 were clear (although if the purpose is to test CV rather than means, a different type of figure could be better). The raw data file contains the important information needed to be understood (e.g., units, scientific names). Note, however, that hypotheses are not presented, and this is a shortcoming of the manuscript.

Experimental design

This manuscript examines functional trait metrics in forests within a conservation area and a buffer zone (representing disturbed forests). Its aims are to build a database of traits and compare functional trait differences among species in the two forest conditions. Overall, their experimental design is appropriate. The analysis of functional traits follows standard protocols and there are no issues there. It would be helpful to have a map showing the spatial extent of the 28 plots in the reserve and the 44 plots in the buffer zone that were selected. They note that several forest types are in the reserve, and it would be useful to see a map of these forest types with plots overlain onto this map. In addition, there is no sense how close the plots were to each other—an issue for understanding statistical independence.

For the statistical analysis, it is a bit unclear. Were the t-tests conducted on the coefficients of variation or the means? The Pearson analysis has many comparisons, and therefore the p values should probably be evaluated by Bonferroni corrections. Also, what is the purpose of these correlations? The justification for looking at the data in this way (a common method) needs to be described in the Introduction and Methods.
Overall, it is unclear in the manuscript whether the aim is to study the means (as the Figures depict) or the CV (as some of the text and stats suggest). I argue that the lack of a clear hypothesis in the Introduction is the root of this confusion. In fact, the Introduction should have at least two hypotheses: one about the differences in conserved vs. disturbed forests (and state if means or CV comparisons are analyzed) and one about scaling relationships (the correlations).

Validity of the findings

The strength of the manuscript is that it provides useful functional trait data on a large number of species, and the data has been collected according to standard practices. The weakness of the manuscript is its lack of context and motivation. What is missing from the manuscript is any motivation of the Introduction of why species might differ in their functional trait values between natural and disturbed forest sites. The Introduction does a good job at describing the ecology of functional traits and their usefulness, but does not make this extension into why they may be interesting to look at in this comparison. And, there are two types of comparisons that could be made: 1) across all species disturbed vs. natural forests; 2) within species that are found in both forest types. Partly because there is not this kind of motivation in the Introduction, the Discussion does not provide integration of the conceptual purposes of the work. In my opinion, this kind of motivation is necessary for this manuscript to be published.

They measured many traits, but context is needed in the Discussion of which ones are most important for the ecology of these species.

Additional comments

Line 225, “plant functional trait data are”; data are plural

Throughout: Abbreviatins is misspelled. I should be "abbreviations"

·

Basic reporting

• The manuscript is generally well written and organized, with relevant references and appropriate structure. However, long author lists in the references unnecessarily increase the length. I recommend applying 'et al.' after 2–3 authors, where allowed by PeerJ formatting policy.
• Emphasize more strongly in the introduction how the stoichiometric perspective provides added ecological insight beyond traditional trait-based ecology.
• Figure 3 (the correlation matrix) is dense and visually difficult to interpret. Consider increasing the font size, resolution, or grouping traits by plant organ, or offering an interactive or supplementary version.
• Trait abbreviations (e.g., LCN, SLA, SCP) should be clearly defined in figure captions and text. Some instances show typos (e.g., 'ration' instead of 'ratio').

Experimental design

• Lines 126–127+abstract: The authors sampled 62 species from disturbed forests and 43 from conserved forests. Does this reflect increased species richness in disturbed areas, or were different selection criteria used? Please clarify this sampling disparity.
• Provide more clarity on how disturbance was defined and quantified. Are these natural, anthropogenic, or mixed disturbances?
• Clarify how species were defined as 'common' or 'dominant'. Were abundance thresholds or expert judgment used?

Validity of the findings

• The conclusion confirms expected trends: trait variability is generally lower in disturbed forests, and many traits differ significantly. However, this is ecologically intuitive. What novel ecological insight does this study offer? Please clarify the contribution beyond the dataset provision.
• No explanation is given for why certain traits showed significant differences while others did not. It is likely that traits respond differently to disturbance based on their ecological sensitivity. Discussing these trait-level mechanisms would improve the manuscript's interpretive value.

Additional comments

• The Discussion is rather brief, considering the study’s aim to address gaps in trait-based analysis of disturbed forests. The authors should engage more with global trait datasets (e.g., TRY) and relevant synthesis studies to compare observed trends.
• There is a missed opportunity to evaluate why traits from leaves are more commonly studied than bark/stem traits. What does this study contribute by expanding the trait set? Discussing this would enhance the paper's novelty.
• Consider adding implications for forest restoration strategies/policies or ecological modeling efforts using this trait database.

---

## Round 0.2 · Minor Revisions

The revised manuscript is improved; however, Reviewer 2 has made additional request regarding presentation of the Results section. Please consider these requests when revising your manuscript.

Reviewer 2 ·

Basic reporting

This manuscript has improved with a better discussion of the disturbed forest situation, part of which is explained in the text and part of it explained with the addition of Figure 1 and Table S1.

Experimental design

The additions that the authors made to clarify the sampling design and methods are now adequate.

Validity of the findings

The data analysis section (lines 198-215) could still use further clarification. It seems like the difference among means between conserved and disturbed forest is the heart of the analysis should come first. Examining coefficients of variation are a way to see which traits are most variable and thus gain insights on the importance of particular traits, which is a secondary goal. However, now the manuscript is still written with CV being central.
In terms of the functional trait relationships, it seems to me it is less important the number of significant correlations in disturbed vs. conserved forests, but to pose the question: were the relationships that were significant in conserved forests also significant among disturbed forests? In other words, do these forest types behave in the same way? It is currently difficult to assess this question in the manuscript.

I also wonder about variation within species. It looks like from the data file that some species both the conserved and disturbed forest types. Does a species in both vary in its traits in the way predicted by the hypotheses? Also relating to the hypotheses, it is much better now that they are in the Introduction, but the manuscript needs to state whether the hypotheses were supported or not in the Introduction.

I believe that this paper could be much more—describing the ecological patterns among the species and forest types—as the data is present to do so. However, this paper seems to aim to mainly describe the database and the additions of data made, without going into much interpretation. I will leave that to the editor to decide if that is appropriate.

·

Basic reporting

The manuscript provides adequate background on trait approaches and stoichiometry and motivates the gap for disturbed forests. This study provides a meaningful comparison of plant functional traits between disturbed and conserved forests, with clear ecological implications. Trait and stoichiometry dataset across multiple organs (leaf, twig, bark, stem) in an ecologically important region is valuable and within scope. I recommend acceptance.

Experimental design

The research question is clearly defined and addresses a meaningful gap. The study design is robust, with a clear sampling strategy.

Validity of the findings

The statistical methods (t-tests, Pearson correlations) are appropriate and well-applied. The data are robust and support the conclusions.

---

## Round 0.3 · accepted · Accept

Thank you for your response to the reviewer. I understand that you plan to reserve additional explorations of the dataset for a future analysis. In four instances, the term "ecological stoichiometry" should be replaced with the term "ecological stoichiometry traits".. This change should be made on L 24, L 88, L 226 and L 263.